# Learning Options via Compression

**Yiding Jiang**[*]
Carnegie Mellon University
yidingji@cs.cmu.edu

**Evan Zheran Liu**[*]
Stanford University
evanliu@cs.stanford.edu

**Benjamin Eysenbach**
Carnegie Mellon University
beysenba@cs.cmu.edu

**J. Zico Kolter**
Carnegie Mellon University
zkolter@cs.cmu.edu

**Chelsea Finn**
Stanford University
cbfinn@cs.stanford.edu

## Abstract

Identifying statistical regularities in solutions to some tasks in multi-task reinforcement learning can accelerate the learning of new tasks. Skill learning offers one way of identifying these regularities by decomposing pre-collected experiences into a sequence of skills. A popular approach to skill learning is maximizing the likelihood of the pre-collected experience with latent variable models, where the latent variables represent the skills. However, there are often many solutions that maximize the likelihood equally well, including degenerate solutions. To address this underspecification, we propose a new objective that combines the maximum likelihood objective with a penalty on the description length of the skills. This penalty incentivizes the skills to maximally extract common structures from the experiences. Empirically, our objective learns skills that solve downstream tasks in fewer samples compared to skills learned from only maximizing likelihood. Further, while most prior works in the offline multi-task setting focus on tasks with low-dimensional observations, our objective can scale to challenging tasks with high-dimensional image observations.

## 1 Introduction

While learning tasks from scratch with reinforcement learning (RL) is often sample inefficient [30, 8], leveraging datasets of pre-collected experience from various tasks can accelerate the learning of new tasks. This experience can be used in numerous ways, including to learn a dynamics model [18, 38], to learn compact representations of observations [87], to learn behavioral priors [77], or to extract meaningful skills or options [79, 42, 2, 59] for hierarchical reinforcement learning (HRL) [6]. Our work studies this last approach. The central idea is to extract commonly occurring behaviors from the pre-collected experience as skills, which can then be used in place of primitive low-level actions to accelerate learning new tasks via planning [70, 51] or RL [55, 47, 52]. For example, in a navigation domain, learned skills may correspond to navigating to different rooms or objects.

Prior methods learn skills by maximizing the likelihood of the pre-collected experience [43, 42, 2, 90]. However, this maximum likelihood objective (or the lower bounds on it) is *underspecified*: it often admits many solutions, only some of which would help learning new tasks (see Figure 1). For example, one degenerate solution is to learn a single skill for each entire trajectory; another degenerate solution is to learn skills that operate for a single timestep, equivalent to the original action space. Both of these decompositions can perfectly reconstruct the pre-collected experiences (and, hence, maximize likelihood), but they are of limited use for learning to solve new tasks. Overall, the maximum likelihood objective cannot distinguish between such decompositions and potentially more useful

---

[*]Equal contribution

36th Conference on Neural Information Processing Systems (NeurIPS 2022).

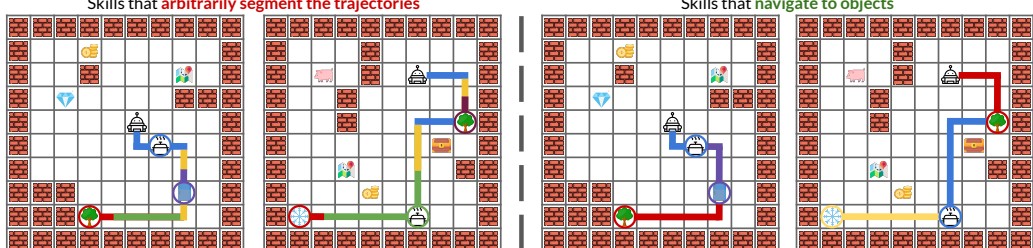

Figure 1: Skill learning via maximizing likelihood is *underspecified*: the skills (represented by the different colored segments) on both the left and the right maximize the likelihood (i.e., encode the data), but the skills on the right are more likely to be useful for a new task. Our approach factors out common temporal structures, yielding the skills on the right. The visualized tasks are from Kipf et al. [42].

decompositions, and we find that this underspecification problem can empirically occur even on simple tasks in our experiments.

To address this underspecification problem, we propose to introduce an additional objective which biases skill learning to acquire skills that can accelerate the learning of new tasks. To construct such an objective, we make two key observations:

- Skills must extract reusable temporally-extended structure from the pre-collected experiences to accelerate learning of new tasks.
- *Compressing* data also requires identifying and extracting common structure.

Hence, we hypothesize that compression is a good objective for skill learning. We formalize this notion of compression with the principle of *minimum description length* (MDL) [69]. Concretely, we combine the maximum likelihood objective (used in prior works) with a new term to minimize the number of bits required to represent the pre-collected experience with skills, which incentivizes the skills to find common structure. Additionally, while prior compression-based methods typically involve discrete optimization and hence are not differentiable, we also provide a method to minimize our compression objective via gradient descent (Section 4), which enables optimizing our objective with neural networks.

Overall, our main contribution is an objective for extracting skills from offline experience, which addresses the underspecification problem with the maximum likelihood objective. We call the resulting approach LOVE (**L**earning **O**ptions **V**ia compr**E**ssion). Using multi-task benchmarks from prior work [42], we find that LOVE can learn skills that enable faster RL and are more semantically meaningful compared to skills learned with prior methods. We also find that LOVE can scale to tasks with high-dimensional image observations, whereas most prior works focus on tasks with low-dimensional observations.

## 2 Related Works

We study the problem of using offline experience from one set of tasks to quickly learn new tasks. While we use the experience to learn skills, prior works also consider other approaches of leveraging the offline experience, including to learn a dynamics model [38], to learn a compact representation of observations [87], and to learn behavioral priors for exploring new tasks [77].

We build on a rich literature on extracting skills [79] from *offline* experience [62, 21, 43, 74, 42, 7, 66, 92, 91, 46, 2, 72, 71, 59, 90, 53, 93, 68, 80, 88]. These works predominantly learn skills using a latent variable model, where the latent variables partition the experience into skills, and the overall model is learned by maximizing (a lower bound on) the likelihood of the experiences. This approach is structurally similar to a long line of work in hierarchical Bayesian modeling [24, 9, 20, 49, 37, 13, 48, 39, 28, 15]. We also follow this approach and build off of VTA [39] for the latent variable model, but differ by introducing a new compression term to address the underspecification problem with maximizing likelihood. Several prior approaches [90, 91, 23] also use compression, but yield open-loop skills, whereas we aim to learn closed-loop skills that can react to the state. Additionally, Zhang et al. [90] show that maximizing likelihood with variational inference can itself be seen as a form of compression, but this work compresses the latent variables at each time step, which does not in general yield optimal compression as our objective does.

Beyond learning skills from offline experience, prior works also consider learning skills with additional supervision [3, 63, 75, 35, 78], from online interaction without reward labels [25, 19, 17, 73, 5], and from online interaction with reward labels [81, 44, 4, 84, 29, 58]. Additionally, a large body of work on meta-reinforcement learning [16, 85, 67, 94, 50] also leverages prior experience to quickly learn new tasks, though not necessarily through skill learning.

## 3   Preliminaries

**Problem setting.**   We consider the problem of using an offline dataset of experience to quickly solve new RL tasks, where each task is identified by a reward function. The offline dataset is a set of trajectories $\mathcal{D} = \{\tau_i\}_{i=1}^{N}$, where each trajectory is a sequence of states $\boldsymbol{x} \in \mathcal{X}$ and actions $\boldsymbol{a} \in \mathcal{A}$: $\tau_i = \{(\boldsymbol{x}_1, \boldsymbol{a}_1), (\boldsymbol{x}_2, \boldsymbol{a}_2), \ldots\}$. Each trajectory is collected by some unknown policy interacting with a Markov decision process (MDP) with dynamics $\mathcal{P}(\boldsymbol{x}_{t+1} \mid \boldsymbol{x}_t, \boldsymbol{a}_t)$. Following prior work [18, 1, 14], we do not assume access to the rewards (i.e., the task) of the offline trajectories.

Using this dataset, our aim is to quickly solve a new task. The new task involves interacting with the same MDP as the data collection policy in the offline dataset, except with a new reward function $\mathcal{R}(\boldsymbol{x}, \boldsymbol{a})$. The objective is to learn a policy that maximizes the expected returns in as few numbers of environment interactions as possible.

**Variational Temporal Abstraction.**   Our method builds upon the graphical model from variational temporal abstraction (VTA) [39], which is a method for decomposing non-control sequential data (i.e., data without actions) into subsequences. We overview VTA below and extend it to handle actions it Section 4.1.

VTA assumes that each observation $\boldsymbol{x}_t$ is associated with an abstract representation $\boldsymbol{s}_t$ and a subsequence descriptor $\boldsymbol{z}_t$. An additional, binary random variable $\boldsymbol{m}_t$ indicates whether a new subsequence begins at the current observation. VTA uses the following generative model:

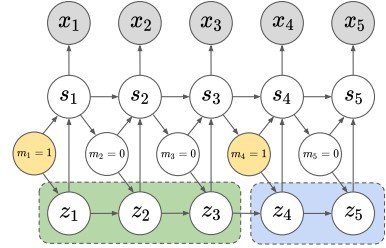

1. Determine whether to begin a new subsequence: $\boldsymbol{m}_t \sim p(\boldsymbol{m}_t \mid \boldsymbol{s}_{t-1})$. A new subsequence always begins on the first time step, i.e., $\boldsymbol{m}_1 = 1$.
2. Sample the descriptor: $\boldsymbol{z}_t \sim p(\boldsymbol{z}_t \mid \boldsymbol{z}_{<t}, \boldsymbol{m}_t)$. If $\boldsymbol{m}_t = 0$, the previous descriptor is copied, i.e., $\boldsymbol{z}_t = \boldsymbol{z}_{t-1}$.
3. Sample the next state abstract representation: $\boldsymbol{s}_t \sim p(\boldsymbol{s}_t \mid \boldsymbol{s}_{<t}, \boldsymbol{z}_t, \boldsymbol{m}_t)$.
4. Sample the observation: $\boldsymbol{x}_t \sim p(\boldsymbol{x}_t \mid \boldsymbol{s}_t)$.

Figure 2: The VTA graphical model (adapted from Kim et al. [39]). The model decomposes the data $x_{1:T}$ into subsequences demarcated by when the boundary variables $m_t = 1$ (highlighted in yellow). Each subsequence is assigned a descriptor $\boldsymbol{z}$, where all $\boldsymbol{z}_t$ within a subsequence are the same (outlined in dashed lines).

We visualize this sampling procedure in Figure 2. Formally, we can write this generative model as:

$$p(\boldsymbol{x}_{1:T}, \boldsymbol{z}_{1:T}, \boldsymbol{s}_{1:T}, \boldsymbol{m}_{1:T}) = \prod_{t=1}^{T} p(\boldsymbol{x}_t \mid \boldsymbol{s}_t)p(\boldsymbol{m}_t \mid \boldsymbol{s}_{t-1})p(\boldsymbol{s}_t \mid \boldsymbol{s}_{<t}, \boldsymbol{z}_t, \boldsymbol{m}_t)p(\boldsymbol{z}_t \mid \boldsymbol{z}_{<t}, \boldsymbol{m}_t).$$

VTA maximizes the likelihood of the observed data under this latent variable model using the standard evidence lower bound (ELBO):

$$\log p(\boldsymbol{x}_{1:T}) \geq \sum_{\substack{\boldsymbol{m}_{1:T} \\ \boldsymbol{z}_{1:T}}} \int_{\substack{\boldsymbol{s}_{1:T}, \\ \boldsymbol{z}_{1:T}}} q_{\boldsymbol{\theta}}(\boldsymbol{z}_{1:T}, \boldsymbol{s}_{1:T}, \boldsymbol{m}_{1:T} \mid \boldsymbol{x}_{1:T}) \cdot \log \frac{p_{\boldsymbol{\theta}}(\boldsymbol{x}_{1:T}, \boldsymbol{z}_{1:T}, \boldsymbol{s}_{1:T}, \boldsymbol{m}_{1:T})}{q_{\boldsymbol{\theta}}(\boldsymbol{z}_{1:T}, \boldsymbol{s}_{1:T}, \boldsymbol{m}_{1:T} \mid \boldsymbol{x}_{1:T})}$$

This lower bound holds for any choice of $q_{\boldsymbol{\theta}}$. VTA chooses to factor this distribution as:

$$q_{\boldsymbol{\theta}}(\boldsymbol{z}_{1:T}, \boldsymbol{s}_{1:T}, \boldsymbol{m}_{1:T} \mid \boldsymbol{x}_{1:T}) = \prod_{t=1}^{T} q_{\boldsymbol{\theta}}(\boldsymbol{m}_t \mid \boldsymbol{x}_{1:t}) \, q_{\boldsymbol{\theta}}(\boldsymbol{z}_t \mid \boldsymbol{z}_{t-1}, \boldsymbol{m}_t, \boldsymbol{x}_{1:T}) \, q_{\boldsymbol{\theta}}(\boldsymbol{s}_t \mid \boldsymbol{z}_t, \boldsymbol{m}_t, \boldsymbol{x}_{1:t}).$$

As discussed above and as we observe in Section 6, the maximum likelihood objective is under-specified and may yield degenerate or unhelpful solutions, which we address in the next section. Specifically, we use the graphical model from VTA, but introduce a new objective that compresses the latent variables $\boldsymbol{z}$, while also maximizing the ELBO.

# 4 LOVE: Learning Options via Compression

In this section, we describe our method for learning skills from pre-collected experience. We first extend the VTA graphical model to handle experience labeled with actions, and then introduce a compression objective that encourages extracting common structure from the experience.

## 4.1 A Graphical Model for Interaction Data

We now extend the VTA graphical model [39] to handle sequential data labeled with actions, where descriptors $z$ now represent skills. From a high level, the model partitions a trajectory into subsequences with the boundary variables $m$ and labels each subsequence as a skill $z$.

We wish to learn a state-conditional policy rather than the joint distribution of the whole trajectory. To do this, we write a new generative model for the actions $a_{1:T}$ conditional on the state $x_{1:T}$:

$$p(z_{1:T}, s_{1:T}, m_{1:T}, a_{1:T} \mid x_{1:T}) = \prod_{t=1}^{T} p(a_t \mid s_t)p(m_t \mid s_{t-1})p(s_t \mid x_t, z_t)p(z_t \mid x_t, z_{t-1}, m_{t-1}).$$

This differs from the original VTA generative model in two ways: (1) we introduce a $p(a_t \mid s_t)$ term that indicates how actions are sampled; and (2) the distributions over $z_t$ and $s_t$ do not depend on all previous $z_{1:t}$ and $s_{1:t}$ to encode the Markov property. We then augment the variational distribution such that the posterior over skills $z$ also depends on actions:

$$q_{\theta}(z_{1:T}, s_{1:T}, m_{1:T} \mid x_{1:T}, a_{1:T}) = \prod_{t=1}^{T} \underbrace{q_{\theta}(m_t \mid x_{1:t})}_{\substack{\text{termination} \\ \text{policy}}} \underbrace{q_{\theta}(z_t \mid z_{t-1}, m_t, x_{1:T}, a_{1:T})}_{\text{skill posterior}} \underbrace{q_{\theta}(s_t \mid z_t, x_t)}_{\text{state abstraction posterior}}.$$

Overall, this yields a model with 3 learned components:

- A *state abstraction posterior* $q_{\theta}(s_t \mid z_t, x_t)$ and *decoder* $p_{\theta}(a_t \mid s_t)$, which together form a distribution over actions conditioned on the skill $z_t$ and current state $x_t$. We refer to these jointly as the *skill policy* $\pi_{\theta}(a_t \mid z_t, x_t) = \mathbb{E}_{s_t \sim q_{\theta}(s_t \mid z_t, x_t)}[p_{\theta}(a_t \mid s_t)]$.
- A *termination policy* $q_{\theta}(m_t \mid x_{1:t})$, which determines whether the previous skill terminates.
- A *skill posterior* $q_{\theta}(z_t \mid z_{t-1}, m_t, x_{1:T}, a_{1:T})$, which outputs the current skill $z_t$ conditioned on all states $x_{1:T}$ and actions $a_{1:T}$. This depends on $m_t$ and $z_{t-1}$, since the boundary variables determine whether the previous skill $z_{t-1}$ terminates: when $m_t = 0$, the previous skill does not terminate and $z_t = z_{t-1}$.

Once this model is learned, the skill policy and termination policy represent the skills, without a need for the skill posterior: Given a skill variable $z$, the skill policy encodes what actions the skill takes, and the termination policy determines when the skill stops. In the next section, we describe our objective for learning this model. Then, in Section 5, we describe how we use the skill policy and termination policy to learn new tasks.

## 4.2 Discovering Structure via Compression

As previously discussed, the maximum likelihood objective is underspecified for skill learning, because many skills can maximize likelihood, independent of whether they are useful for learning new tasks. In this section, we address this with a new objective that attempts to measure how useful skills will be for learning new tasks in terms of compression. Our objective is based on the intuition that effectively compressing a sequence of data requires factoring out common structure, and factoring out common structure is critical for learning useful skills.

We measure the complexity of a skill decomposition as the amount of information required to communicate the sequence of skills that encode the pre-collected experience. The latent variable model introduced in the previous section encodes each trajectory as a sequence of skills $z_{1:T}$ and boundaries $m_{1:T}$. However, using the skill and termination policies, it is possible to recover each trajectory from only the skill variables at the boundary points: i.e., at the time steps $\{t \in [T] \mid m_t = 1\}$. For a prior on skills $p_z$, an optimal code requires $-\log p_z(z_t)$ bits to send a skill $z_t$ [12, Chapter 5.2]. Hence,

the expected code length of communicating a trajectory $\boldsymbol{\tau}_{1:T} = \{(\boldsymbol{x}_1, \boldsymbol{a}_1), \ldots, (\boldsymbol{x}_T, \boldsymbol{a}_T)\}$ is:

$$\text{INFOCOST}(\boldsymbol{\theta}; p_{\boldsymbol{z}}) = -\mathbb{E}_{\substack{\boldsymbol{\tau}_{1:T}, \\ \boldsymbol{m}_{1:T}, \\ \boldsymbol{z}_{1:T}}} \left[ \sum_{t=1}^{T} \log p_{\boldsymbol{z}}(\boldsymbol{z}_t) \boldsymbol{m}_t \right],$$

where the expectation is under trajectories $\boldsymbol{\tau}_{1:T}$ from the pre-collected experience and sampling $\boldsymbol{z}_{1:T}$ from the skill posterior and $\boldsymbol{m}_{1:T}$ from the termination policy. The choice of prior that minimizes the average code length is one that equals the empirical distribution of skills under the pre-collected experience [12, Chapter 5.3]:

$$p_{\boldsymbol{z}}^{\star} \triangleq \arg\min_{p_{\boldsymbol{z}}} \text{INFOCOST}(\boldsymbol{\theta}; p_{\boldsymbol{z}}), \quad \text{where } p_{\boldsymbol{z}}^{\star}(\boldsymbol{z}) = \frac{1}{n_{\text{s}}} \mathbb{E}_{\substack{\boldsymbol{\tau}_{1:T}, \\ \boldsymbol{m}_{1:T}, \\ \boldsymbol{z}_{1:T}}} \left[ \sum_{t=1}^{T} \delta(\boldsymbol{z}_t = \boldsymbol{z}) \boldsymbol{m}_t \right],$$

and $n_{\text{s}} \triangleq \mathbb{E}_{\boldsymbol{\tau}_{1:T}, \boldsymbol{m}_{1:T}} \left[ \sum_{t=1}^{T} \boldsymbol{m}_t \right]$. In Appendix A.1, we show that this marginal $p_{\boldsymbol{z}}^{\star}$ is a proper density for both continuous and discrete $\boldsymbol{z}$. Substituting in this optimal choice of prior, we can show that the code length can be expressed as the marginal entropy $\mathcal{H}_{p_{\boldsymbol{z}}^{\star}}[\boldsymbol{z}]$ times the number of skills per trajectory $n_{\text{s}}$ (see Appendix A.2 for proof):

$$\mathcal{L}_{\text{CL}}(\boldsymbol{\theta}) \triangleq \min_{p(\boldsymbol{z}_t)} \text{INFOCOST}(\boldsymbol{\theta}; p(\boldsymbol{z}_t)) = -\mathbb{E}_{\substack{\boldsymbol{\tau}_{1:T}, \\ \boldsymbol{m}_{1:T}, \\ \boldsymbol{z}_{1:T}}} \left[ \sum_{t=1}^{T} \log p_{\boldsymbol{z}}^{\star}(\boldsymbol{z}_t) \boldsymbol{m}_t \right] = n_{\text{s}} \mathcal{H}_{p_{\boldsymbol{z}}^{\star}}[\boldsymbol{z}]. \quad (1)$$

Intuitively, this is equal to the average code length of a skill multiplied by the the average number of skills per trajectory. Note that compression is only beneficial if the model also achieves high likelihood of the data. We capture this by solving the following constrained optimization problem:

$$\min_{\boldsymbol{\theta}} \ \mathcal{L}_{\text{CL}}(\boldsymbol{\theta}) \quad \text{s.t.} \ \mathcal{L}_{\text{ELBO}}(\boldsymbol{\theta}) \leq C, \quad (2)$$

where $\mathcal{L}_{\text{ELBO}}(\boldsymbol{\theta})$ a negated evidence lower bound on the log-likelihood (detailed in Appendix B).

**Remarks.** We discuss a connection between this objective and variational inference in Appendix A.3. Additionally, while this work focuses on the RL setting, our objective generally applies to sequential modeling problems. We believe that it could be useful for many applications beyond option learning.

### 4.3 Connections to Minimal Description Length

Our approach closely relates to the minimum description length (MDL) principle [69]. This principle equates *learning* with *finding regularity* in data, which can be used to *compress* the data. Informally, the best model is the one that encodes the data with the lowest description length. Given a model $\boldsymbol{\theta}$ that encodes the data $\mathcal{D}$ into some message, one way to formalize the description length of the data $L(\mathcal{D})$ is with a crude two-part code [26]. This decomposes the description length of the data as the length of the message plus the length of the model:

$$L(\mathcal{D}) = \underbrace{L(\mathcal{D} \mid \boldsymbol{\theta})}_{\text{message length}} + \underbrace{L(\boldsymbol{\theta})}_{\text{model length}}. \quad (3)$$

In our case, the message is the latent variables $\boldsymbol{z}_t$ at the boundary points $\{t \in [T] \mid \boldsymbol{m}_t = 1\}$, which can be decoded into the data $\mathcal{D}$ with the skill and termination policies (representing the model). Hence, our approach can be seen as an instance of minimizing the description length $L(\mathcal{D})$. Optimizing our objective in Equation 1 is equivalent to minimizing the message length $L(\mathcal{D} \mid \boldsymbol{\theta})$. While we do not directly attempt to minimize the model length term $L(\boldsymbol{\theta})$, many works [60, 82] indicate that deep learning has *implicit regularization*, which biases optimization toward low complexity solutions without an explicit regularizer. In general, computing the true description length or appropriate notion of complexity for neural networks is a tall order [61, 89, 36]; however, there is a rich space of methods to be explored here and we therefore leave the extension of our approach to directly minimizing the model length term for future work.

### 4.4 A Practical Implementation

**Model.** We instantiate our model by defining discrete skills $\boldsymbol{z} \in [K]$, state abstractions $\boldsymbol{s} \in \mathbb{R}^d$, and binary boundary variables $\boldsymbol{m} \in \{0, 1\}$. We parameterize all components of our model as neural networks. See Appendix D for architecture details.

**Optimization.**    We apply Gumbel-softmax [34, 54] to optimize over the discrete random variables $z$ and $m$. We rewrite the compression objective $\mathcal{L}_{\mathrm{CL}}$ as a product of $n_{\mathrm{s}}$ and $\mathcal{H}_{p_z^\star}[z_t]$ in Equation 1 to improve the stability of optimization. This rewriting allows computing a gradient in terms of a distribution over the skill variables $p_z^\star$, rather than samples $z_t$, which yields a more accurate finite sample gradient. Empirically, this leads to stabler optimization and convergence to better solutions.

We approximate the optimal skill prior $p_z^\star$ using our skill posterior as:

$$p_z^\star(z) \approx \frac{1}{\widehat{\mathbb{E}}_{\substack{\boldsymbol{\tau}_{1:T}, \\ \boldsymbol{m}_{1:T}}}\left[\sum_{t=1}^T \boldsymbol{m}_t\right]} \widehat{\mathbb{E}}_{\substack{\boldsymbol{\tau}_{1:T}, \\ \boldsymbol{m}_{1:T}, \\ \boldsymbol{z}_{1:T}}}\left[\sum_{t=1}^T q_{\boldsymbol{\theta}}(\boldsymbol{z}_t = z \mid \boldsymbol{z}_{t-1}, \boldsymbol{m}_t, \boldsymbol{x}_{1:T}, \boldsymbol{a}_{1:T})\,\boldsymbol{m}_t\right],$$

where $\widehat{\mathbb{E}}$ denotes the empirical expectation over minibatches of $\boldsymbol{x}_{1:T}, \boldsymbol{a}_{1:T}$ from $\mathcal{D}$ and sampling $\boldsymbol{m}_{1:T}$ from the termination policy. We solve the constrained optimization problem (Equation 2) with dual gradient descent on the Lagrangian. In Appendix G, we summarize the overall training procedure in Algorithm 1 and report details about the Lagrangian.

**Enforcing a minimum skill length.**    Though degenerate solutions such as skills that only take a single action score poorly in our compression objective, empirically, such solutions create local optima that are difficult to escape. To avoid this, we mask the boundary variables during training to ensure that each skill $z_t$ operates for at least $T_{\mathrm{min}} = 3$ time steps. We find that when the skills are at least minimally temporally extended, optimization of the compression objective appears to be stabler and achieves better values. We remove these masks at test time, when we learn a task with our skills.

## 5   Using the Learned Skills for Hierarchical RL

We now describe how we quickly learn new tasks, given the skills learned from the pre-collected experience. Overall, we simply augment the agent's action space with the learned skills and learn a new policy that can take either low-level actions or learned skills, similar to Kipf et al. [42]. However, our compression objective may result in unused skills, since this decreases the encoding cost of the used skills. Therefore, we first filter down to only the skills that are used to compress the pre-collected experience by selecting the skills where the marginal is over some threshold $\alpha$:

$$\mathcal{Z} = \left\{ k \in [K] \;\mid\; p_z^\star(k) > \alpha \right\}.$$

Then, on a new task with action space $\mathcal{A}$, we train an agent using the augmented action space $\mathcal{A}^+ = \mathcal{A} \cup \mathcal{Z}$. When the agent selects a skill $z \in \mathcal{Z}$, we follow the procedure in Algorithm 2 in the Appendix. We take actions following the skill policy $\pi_{\boldsymbol{\theta}}(\boldsymbol{a}_t \mid \boldsymbol{z}, \boldsymbol{x}_t)$ (lines 2–3), until the termination policy $q_{\boldsymbol{\theta}}(\boldsymbol{m}_t \mid \boldsymbol{x}_{1:t})$ outputs a termination $\boldsymbol{m}_t = 1$ (line 5). At that point, the agent observes the next state $\boldsymbol{x}_{t+1}$ and selects the next action or skill.

## 6   A Didactic Experiment: Frame Prediction

Before studying the RL setting, we first illustrate the effect of LOVE in the simpler setting of sequential data without actions. In this setting, we use the original VTA model, which partitions a sequence $\boldsymbol{x}_{1:T}$ into contiguous subsequences with the boundary variables $\boldsymbol{m}$ and labels each subsequence with a descriptor $\boldsymbol{z}$, as described in Section 3. Here, VTA's objective is to maximize the likelihood of the sequence $\boldsymbol{x}_{1:T}$, and our compression objective applies to this setting completely unmodified. We compare LOVE and only maximizing likelihood, i.e., VTA [39], by measuring how well the learned subsequences correspond to underlying patterns in the data. To isolate the effect of the compression objective, we do not enforce a minimum skill length.

**Dataset.**    We consider two simple datasets *Simple Colors* and *Conditional Colors*, which consist of sequences of $32 \times 32$ monochromatic frames, with repeating underlying patterns. *Simple Colors* consists of 3 patterns: 3 consecutive yellow frames, 3 consecutive blue frames and 3 consecutive green frames. The dataset is generated by sampling these patterns with probability 0.4, 0.4, and 0.2 respectively and concatenating the results. Each pattern is sampled independent of history. The dataset is then divided into sequences of 6 patterns, equal to $6 \times 3 = 18$ frames.

In contrast to *Simple Colors*, *Conditional Colors* tests learning patterns that depend on history. It consists of 4 patterns: the 3 patterns from *Simple Colors* with an additional pattern of 3 consecutive

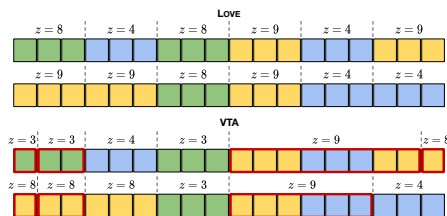

Figure 3: Learned boundaries and descriptors on *Simple Colors*. LOVE recovers the patterns, while VTA learns boundaries that break up patterns or span multiple patterns (outlined in red).

Table 1: Effect of LOVE on the *Simple Colors* and *Conditional Colors* datasets (5 seeds). All approaches achieve similar values for the likelihood objective, but LOVE better recovers the correct boundaries.

|  | *Simple Colors* | | *Conditional Colors* | |
|  | **VTA** | **LOVE** | **VTA** | **LOVE** |
|---|---|---|---|---|
| $\mathcal{L}_{\text{ELBO}}$ | $2868 \pm 43$ | $2838 \pm 19$ | $2832 \pm 7.7$ | $2827 \pm 1.9$ |
| Precision | $0.87 \pm 0.19$ | $\mathbf{0.99} \pm 0.01$ | $0.84 \pm 0.22$ | $\mathbf{0.99} \pm 0.01$ |
| Recall | $0.79 \pm 0.13$ | $\mathbf{0.85} \pm 0.03$ | $0.82 \pm 0.16$ | $\mathbf{0.83} \pm 0.06$ |
| F1 | $0.82 \pm 0.13$ | $\mathbf{0.91} \pm 0.02$ | $0.83 \pm 0.19$ | $\mathbf{0.90} \pm 0.03$ |
| Code Length | $7.58 \pm 1.3$ | $\mathbf{6.34} \pm 0.75$ | $9.17 \pm 1.76$ | $\mathbf{6.83} \pm 0.51$ |

purple frames. The dataset is generated by uniformly sampling the patterns from *Simple Colors*. To make the current timestep pattern dependent on the previous timestep, the yellow frames are re-colored to purple, if the previous pattern was blue or yellow. As in *Simple Colors*, the dataset is divided into sequences of 6 patterns.

In both datasets, the optimal encoding strategy is to learn 3 subsequence descriptors (i.e., skills), one for each pattern in *Simple Colors*. In *Conditional Colors*, the descriptor corresponding to yellow in *Simple Colors* either outputs yellow or purple, depending on the history. This encoding strategy achieves an expected code length of 6.32 nats in *Simple Colors* and 6.59 nats in *Conditional Colors*.

**Results.** We visualize the learned descriptors and boundaries in Figure 3. LOVE successfully segments the sequences into the patterns and assigns a consistent descriptor $z$ to each pattern. In contrast, despite the simple structure of this problem, VTA learns subsequences that span over multiple patterns or last for fewer timesteps than a single pattern.

Quantitatively, we measure the (1) precision, recall, and F1 scores of the boundary prediction, (2) the ELBO of the maximum likelihood objective and (3) the average code length $\mathcal{L}_{\text{CL}}$ in Table 1. While both VTA and LOVE achieve approximately the same negated ELBO (lower is better), LOVE recovers the correct boundaries with much higher precision, recall, and F1 scores. This illustrates that the underspecification problem can occur even in simple sequential data. Additionally, we find that LOVE achieves an encoding cost close to the optimal value. One interesting, though rare, failure mode is that LOVE sometimes even achieves a lower encoding cost than the optimal value by over-weighting the compression term and imperfectly reconstructing the data. In Appendix C, we conduct a ablation study on the weights of $\mathcal{L}_{\text{CL}}$ in optimization.

## 7 Experiments

We aim to answer three questions: (1) Does LOVE learn semantically meaningful skills? (2) Do skills learned with LOVE accelerate learning for new tasks? (3) Can LOVE scale to high-dim pixel observations? To answer these questions, we learn skills from demonstrations and use these skills to learn new tasks, first on a 2D multi-task domain, and later on a 3D variant.

**Multi-task domain.** We consider the multi-task $10 \times 10$ grid world introduced by Kipf et al. [42], a representative skill learning approach (Figure 4). This domain features a challenging control problem that requires the agent to collect up to 5 objects, and we later add a perception challenge to it with the 3D variant. We use our own custom implementation, since the code from Kipf et al. [42] is not publicly available.

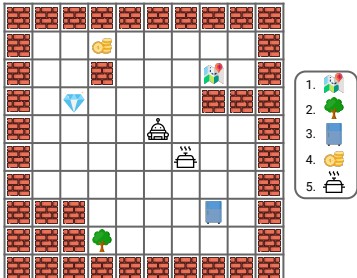

Figure 4: Multi-task grid world instance from Kipf et al. [42]. The agent must pick up a sequence of objects in a specified order.

In each task, 6 objects are randomly sampled from a set of $N_{\text{obj}} = 10$ different objects and placed at random positions in the grid. Additionally, impassable walls are randomly added to the grid, and the agent is also placed at a random initial grid cell. Each task also includes an instruction list of $N_{\text{pick}} = 3$ or 5 objects that the agent must pick up in order.

Within each task, the agent's actions are to move up, down, left, right, or to pick up an item at the agent's grid cell. The state consists of two parts: (1) a $10 \times 10 \times (N_{\text{obj}} + 2)$ grid observation,

indicating if the agent, a wall, or any of the $N_{\text{obj}}$ different object types is present at each of the $10 \times 10$ grid cells; (2) the instruction corresponding to the next object to pick up, encoded as a one-hot integer. Following Kipf et al. [42], we consider two reward variants: In the *sparse reward* variant, the agent only receives $+1$ reward after picking up *all* $N_{\text{pick}}$ objects in the correct order. In the *dense reward* variant, the agent receives $+1$ reward after picking up each specified object in the correct order. Our dense reward variant is slightly harder than the variant in Kipf et al. [42], which gives the agent $+1$ reward for picking up objects in *any* order. The agent receives $0$ reward in all other time steps. The episode ends when the agent has picked up all the objects in the correct order, or after $50$ timesteps.

**Pre-collected experience.** We follow the setting in Kipf et al. [42]. We set the pre-collected experience to be $2000$ demonstrations generated via breadth-first search on randomly generated tasks with only $N_{\text{pick}} = 3$ and test if the agent can generalize to $N_{\text{pick}} = 5$ when learning a new task. These demonstrations are not labeled with rewards and also *do not* contain the instruction list observations.

**Points of comparison.** To study our compression objective, we compare with two representatives of learning skills via the maximum likelihood objective:

- VTA [39], modified to handle interaction data as in Section 4.1.
- Discovery of deep options (DDO) [21]. We implement DDO's maximum likelihood objective and graphical model on top of VTA's variational inference optimization procedure, instead of expectation-gradients used in the original paper.

For fairness, we also compare with variants of these that implement the minimum skill length constraint from Section 4.4. CompILE [42] is another approach that learns skill by maximizing likelihood and was introduced in the same paper as this grid world. Because the implementation is unavailable, we do not compare with it. However, we note that CompILE requires additional supervision that LOVE, VTA, and DDO do not: namely, it requires knowing how many skills each demonstration is composed of. This supervision can be challenging to obtain without already knowing what the skills should be.

Since latent variable models are prone to local optima [45], it is common to learn such models with multiple restarts [57]. We therefore run each method with 3 random initializations and pick the best model according to the compression objective $\mathcal{L}_{\text{CL}}$ for LOVE and according to the ELBO of the maximum likelihood objective $\mathcal{L}_{\text{ELBO}}$ for the others. Notably, this does not require any additional demonstrations or experience.

We begin by analyzing the skills learned from the pre-collected experience before analyzing the utility of these skills for learning downstream tasks in the next sections.

## 7.1 Analyzing the Learned Skills

We analyze the learned skills by comparing them to a natural decomposition that partitions the demonstration in $N_{\text{pick}}$ sequences of moving to and picking up an object. Specifically, we measure the precision and recall of each method in outputting the correct boundaries of these $N_{\text{pick}}$ sequences.

We find that only LOVE recovers the correct boundaries with both high precision and recall (Table 2). Visually examining the

Table 2: Precision and recall of outputting the boundaries between picking up different objects. Only LOVE learns skills that move to and pick up an object.

|  | Precision | Recall | F1 |
|---|---|---|---|
| DDO [21] | 0.19 | **1** | 0.32 |
| DDO + min. skill length | 0.26 | 0.53 | 0.35 |
| VTA [39] | 0.19 | **0.99** | 0.32 |
| VTA + min. skill length | 0.27 | 0.53 | 0.36 |
| LOVE (ours) | **0.90** | 0.94 | **0.92** |

skills learned with LOVE shows that each skill moves to an object and picks it up. We visualize LOVE's skills in Appendix H. In contrast, maximizing likelihood with either DDO or VTA learns degenerate skills that output only a single action, leading to high recall, but low precision. Adding the minimum skill length constraint to these approaches prevents such degenerate solutions, but still does not lead to the correct boundaries. Skills learned by VTA with the minimum skill length constraint exhibit no apparent structure, but skills learned by DDO with the minimum skill length constraint appear to move toward objects, though they terminate at seemingly random places. These results further illustrate the underspecification problem. All approaches successfully learn to reconstruct the

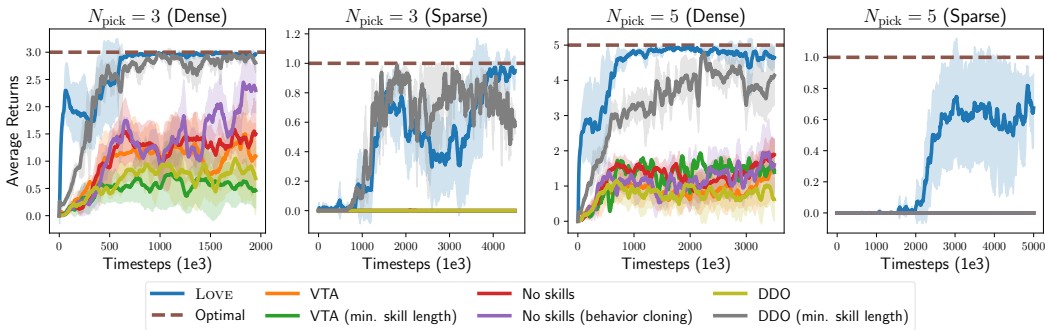

Figure 5: **Sample efficient learning.** We plot returns vs. timesteps of environment interactions for 4 settings in the grid world with 1-stddev error bars (5 seeds). Only LOVE achieves high returns across all 4 settings.

demonstrations and achieve high likelihood, but only LOVE and DDO with the minimum skill length constraint learn skills that appear to be semantically meaningful.

## 7.2 Learning New Tasks

Next, we evaluate the utility of the skills for learning new tasks. To evaluate the skills, we sample a new task in each of 4 settings: with sparse or dense rewards and with $N_{pick} = 3$ or $N_{pick} = 5$. Then we learn the new tasks following the procedure in Section 5: we augment the action space with the skills and train an agent over the augmented action space. To understand the impact of skills, we also compare with a low-level policy that learns these tasks using only the original action space, and the same low-level policy that incorporates the demonstrations by pre-training with behavioral cloning. We parametrize the policy for all approaches with dueling double deep Q-networks [56, 86, 83] with $\epsilon$-greedy exploration. We report the returns of evaluation episodes with $\epsilon = 0$, which are run every 10 episodes. We report returns averaged over 5 seeds. See Appendix F.4 for full details.

**Results.**    Overall, LOVE learns new tasks across all 4 settings comparably to or faster than both skill methods based on maximizing likelihood and incorporating demonstrations via behavior cloning (Figure 5). More specifically, when $N_{pick} = 3$, DDO with the min. skill length constraint performs comparably to LOVE, despite not segmenting the demonstrations into sequences of moving to and picking up objects. We observe that while a single DDO skill does not move to and pick up an object, the skills consistently move toward objects, which likely helps with exploration. Imitating the demonstrations with behavior cloning also accelerates learning when $N_{pick} = 3$ with dense rewards, though not as much as LOVE or DDO, and yields insignificant benefit in all other settings. Skill learning with VTA yields no benefit over no skill learning at all.

Recall that $N_{pick} = 3$ in all of the demonstrations. LOVE learns new tasks in the generalization setting of $N_{pick} = 5$ much faster than the other methods. With dense rewards, DDO (min. skill length) also eventually solves the task, but requires over $8\times$ more timesteps. The other approaches learn to pick up some objects, but never achieve optimal returns of 5. With sparse rewards, LOVE is the only approach that achieves high returns, while all other approaches achieve 0 returns. This likely occurs because this setting creates a challenging exploration problem, so exploring with low-level actions or skills learned with VTA or DDO rarely achieves reward. By contrast, exploring with skills learned with LOVE achieves rewards much more frequently, which enables LOVE to ultimately solve the task.

## 7.3 Scaling to High-dimensional Observations

Prior works in the offline multi-task setting have learned skills from low-dimensional states and then transferred them to high-dim pixel observations [68], or have learned hierarchical models from pixel observations [39, 21]. However, to the best of our knowledge, prior works in this setting have not directly learned skills from high-dim pixel observations, only from low-dimensional states. Hence, in this section, we test if LOVE can scale up to high-dim pixel observations by considering a challenging 3D image-based variant of the multi-task domain above, illustrated in Figure 6. The goal is still to pick up $N_{pick}$ objects in the correct order, where objects are blocks of different colors. Each task includes 3 objects randomly sampled from a set of $N_{obj} = 6$ different colors. The observations are $400 \times 60 \times 3$ egocentric panoramic RGB arrays. The actions are to move forward / backward, turn left / right, and pick up.

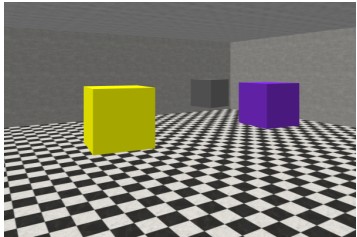

Figure 6: 3D visual multi-task domain.

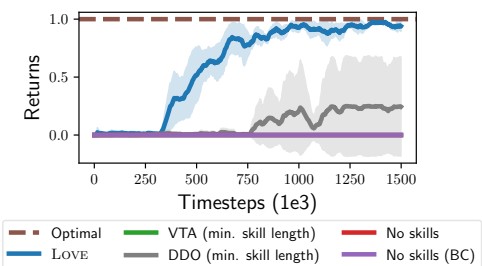

Figure 7: Returns on the 3D visual multi-task domain with 1-stddev error bars (5 seeds). Only LOVE achieves near-optimal returns.

We consider only the harder *sparse reward* and generalization setting, where the agent must pick up $N_{pick} = 4$ objects in the correct order, after receiving 2,000 pre-collected demonstrations of the approximate shortest path of picking up $N_{pick} = 2$ objects in the correct order. We use the same hyperparameters as in multi-task domain and only change the observation encoder and action decoder (full details in Appendix D.3). Figure 7 shows the results. Again, LOVE learns skills that each navigate to and pick up an object, which quickly achieves optimal returns, indicating its ability to scale to high-dim observations. In contrast, the other approaches perform much worse on pixel observations than the previous low-dim observations. DDO also makes progress on the task, but requires far more samples, as it again learns skills that only operate for a few timesteps, though they move toward the objects. All other approaches fail to learn at all, including the variants of VTA and DDO without the min. skill length, which are not plotted.

## 8 Conclusion

We started by highlighting the underspecification problem: maximizing likelihood does not necessarily yield skills that are useful for learning new tasks. To overcome this problem, we drew a connection between skill learning and compression and proposed a new objective for skill learning via compression and a differentiable model for optimizing our objective. Empirically, we found that the underspecification problem occurs even on simple tasks and learning skills with our objective allows learning new tasks much faster than learning skills by maximizing likelihood.

Still, important future work remains. LOVE applies when there are useful and consistent structures that can be extracted from multiple trajectories. This is often present in multi-task demonstrations, which solve related tasks in similar ways. However, an open challenge for adapting to general offline data like D4RL [22], is to ensure that the learned skills do not overfit to potentially noisy or unhelpful behaviors often present in offline data. In addition, we showed a connection between skill learning and compression by minimizing the description length of a crude two-part code. However, we only proposed a way to optimize the message length term and not the model length. Completing this connection by accounting for model length may therefore be an promising direction for future work.

**Reproducibility.** Our code is publicly available at https://github.com/yidingjiang/love.

**Acknowledgements.** We would like to thank Karol Hausman, Abhishek Gupta, Archit Sharma, Sergey Levine, Annie Xie, Zhe Dong, Samuel Sokota for valuable discussions during the course of this work, and Victor Akinwande, Christina Baek, Swaminathan Gurumurthy for comments on the draft. We would also like to thank the anonymous reviewers for their valuable feedback. Last, but certainly not least, YJ and EZL thank the beautiful beach of Kona for initiating this project.

YJ is supported by funding from the Bosch Center for Artificial Intelligence. EZL is supported by a National Science Foundation Graduate Research Fellowship under Grant No. DGE-1656518. CF is a Fellow in the CIFAR Learning in Machines and Brains Program. This work was also supported in part by the ONR grant N00014-21-1-2685, Google, and Intel. Icons in this work were made by ThoseIcons, FreePik, VectorsMarket, from FlatIcon.

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
