# OpenReview forum: "Learning Options via Compression"
_NeurIPS.cc/2022/Conference — NeurIPS 2022 Accept_

### Official Review · Reviewer_gYjp · 2022-06-14

**Rating:** 7
**Confidence:** 3
**Soundness:** 3 good
**Presentation:** 3 good
**Contribution:** 2 fair

**Summary:**

This paper presents a regularization method for learning skills from data. The authors are inspired by the minimum description length and combine the maximum likelihood objective with the compressing term to improve the effectiveness of learned skills. The resulting approach LOVE demonstrates the faster-converging speed and semantically important results compared to prior work. Furthermore, LOVE can scale to tasks with image observations.

**Questions:**

1. Why don't the authors try to experiment on Atari to show the learned skills?
2. Why can LOVE discover better boundaries in Figure 3?
3. How to balance the maximum likelihood target and the length constraint target? If there are many skills of the same length, would the target become ineffective?
4. Can the proposed compresion scheme generalize to other option learning frameworks?

**Limitations:**

The authors shortly describe the limitations in the conclusion part. However, I don't quite understand that part and I think the important limitations of this paper are not discussed (refer to the above limitation part).

**Strengths And Weaknesses:**

## After reading other reviewers' comments which are more positive than I thought and reading other rebuttals, I decided to increase my score from five to six and then to seven.
# Strength
1. Learning skills from demonstrations is an important task in hierarchical reinforcement learning.
2. The proposed compression metric should be effective to improve the learned skills in HRL.
3. The proposed visualizations in Figure3 and Table1 demonstrate the effectiveness of LOVE in various perspectives.
4. The code is provided.

# Weakness
1. The authors state in many places, "prior works do not directly learn skills from high-dimensional pixel observations". This is false. In fact, many work tries to learn skills on Atari games (real-image games), such as the option critic architecture [4] and RAPL [83].
2. The proposed compression term is merely a trick beyond VTA. And the compression metric is already used in RAPL [83] and the unsupervised skill discovery literature. So the novelty is limited.
3.  The title is a bit misleading. The title states "learning options via compression". However, the compression is merely a regression term in the learning scheme, and the major body of this approach is still VTA.

---

> ### Author Response · Authors · 2022-08-02
> **Response to Reviewer gYjp, Part 1/2**
>
> We thank the reviewer for the feedback and questions. We address each point below and have updated the paper accordingly.
>
> 1. We agree with the reviewer that several option-learning algorithms, such as **RAPL and Option-Critic discover options from high-dim pixel observations**. We note that we consider a different setting from these works. We consider learning options from multiple tasks and demos, whereas these works consider learning from a single task. In the multi-task setting, there are far fewer works that learn from pixels. However, we understand that the claim may be misleading and have revised the claims in the abstract and Section 7.3 accordingly. Furthermore, we have added comparison to Option-Critic (more details in Appendix I) and found that LOVE significantly outperforms Option-Critic. This suggests that existing online methods are likely not sufficient for the tasks we consider in the paper.
> |            | N_pick = 3 (Dense) | N_pick = 5 (Sparse) |
> |------------|--------------------|---------------------|
> | Option-Critic | 2.0                | 0.0                 |
> | LOVE (ours) | **3.0**                | **0.7**                 |
>
>     *Maximum returns achieved by Option-Critic and LOVE.*
>
> 2. **Novelty and “the compression term is merely a trick beyond VTA.”** We’d like to clarify that while our work builds off of VTA, the core and novel contribution is observing that skill discovery is underspecified, and addressing this via an instantiation of the MDL principle (a canonical information-theoretic formalism for compression, hence the title), which attempts to learn skills that maximally extract structure from the demos. Additionally, as compression typically involves discrete objects, providing a fully differentiable objective, as we do, requires significant technical innovation (see Section 4.4). This enables us to use function approximators and scalable gradient-based methods to compress trajectories, which is novel – while other works, such as RAPL and [2] also use compression, they compress discrete objects (using classical compression algorithms such as Sequitur) and do not provide differentiable compression objectives as LOVE does. Further, compression used in prior works extracts open-loop skills that do not condition on the state, whereas LOVE yields closed-loop skills that can adapt based on the state. We note that RAPL offers an interesting way to leverage the learned skills, and combining that with LOVE could be promising future work.
>
> 3. **“Why can LOVE discover better boundaries in Figure 3?”** For maximum likelihood methods, like VTA and DDO, the objective does not distinguish between different segmentations/boundaries that achieve the same high likelihood (i.e., the underspecification problem), and in particular, these methods are not incentivized to find the true boundaries. However, here, the best compression is achieved by recovering the generative process of the data, which corresponds to finding the true boundaries. Since LOVE compresses the demos, it finds the true boundaries.

---

> > ### Author Response · Authors · 2022-08-02
> > **Response to Reviewer gYjp, Part 2/2**
> >
> > 4. **“Why don't the authors try to experiment on Atari?”** We consider a multi-task setting, where learning skills on some tasks enables quickly learning new, but related tasks. Since in Atari, there is not a clear set of tasks that are related to each other, we evaluate on benchmarks from prior works in this multi-task setting, specifically the environment from ComPILE, and a harder 3D vision-based variant of the environment.
> >
> > 5. **“How do you balance the maximum likelihood [term] and the [compression term]?”** We do so automatically via dual gradient descent [1] on the constrained optimization problem. We discuss the details of this technique in Appendix G. We will make sure to emphasize this in the revision.
> >
> > 6. **"Can the compression term handle many skills of the same length?"** We are unsure if we fully understand the question. Note that the word length may be overloaded here: in our objective, we refer to the length of a skill as the number of bits it takes to encode. This is the log probability of how often a skill is used, which is unrelated to how many timesteps it lasts. Therefore, the compression objective is not just about increasing or decreasing the duration of the skill — instead, it’s trying to recover the maximal common structure in the data, which is unaffected by whether multiple skills have the same length. For example, in the Simple and Conditional Colors, LOVE recovers the true data boundaries and learns multiple skills, which all last 3 timesteps, without issue. If we misunderstood the question, please let us know and we’d be happy to discuss this further.
> >
> > 7. **"Can you use LOVE on other Option-Learning frameworks?"** This is an excellent question. In principle, LOVE only requires a graphical model that has latent variables $z$ (for the skills) and $m$ (for the boundary), which enable us to measure the coding length. In this work, we consider learning skills from offline demos. In this setting, LOVE applies to any methods that leverage any graphical model that parametrizes boundaries this way. However, there are other settings as well, such as learning skills online (like option-critic), or learning skills from offline data that are not demos. In these cases, we may still use the LOVE compression term as a regularizer for finding structure, but more work needs to be done to figure out how to balance different aspects such as online exploration or noise, and deal with data with less common structure. We believe that these are exciting directions for future works and that LOVE provides an important step towards these goals.
> >
> > **Reference**
> >
> > [1] https://web.stanford.edu/class/ee364b/lectures/primal_dual_subgrad_slides.pdf
> >
> > [2] A Compression-Inspired Framework for Macro Discovery. Garcia et al.

---

> ### Comment · Reviewer_gYjp · 2022-08-03
> **A good rebuttal.**
>
> I think overall the authors did a good rebuttal. My questions are mostly addressed. Let's wait other negative reviewers' opinions.

---

### Official Review · Reviewer_94Fc · 2022-07-11

**Rating:** 6
**Confidence:** 3
**Soundness:** 3 good
**Presentation:** 3 good
**Contribution:** 2 fair

**Summary:**

This paper proposes a framework to discover options from the offline data. The framework is build upon the Variational Temporal Abstraction(VTA) method. To discover semantically meaningful skills, they involve an optimization object of compression. They carry out experiments on 2D and 3D multi-task domains. The experiment results show that the proposed framework is able to discover meaningful options and reuses them for new tasks.

**Questions:**

1. There are a lot of methods for offline option discovery proposed rencently. Can you explain the reason of choosing DDO?

**Limitations:**

The authors have adequately addressed the limitations and potential negative societal impact of their work.

**Strengths And Weaknesses:**

Strengths:
1. The proposed method is reasonable for option discovery.
2. The analysis and introduction of the method is very clear and make it easy to follow.

Weaknesses:
1. Although I'm not an expert in the field of option discovery from offline data, the idea of penalizing option switching or reward option continuation is common is option discovery from online data., e.g. option-critic[1] and [2]. Thus, the novelty may be not enough.
2. The choice of comparison method. There are a lot of methods for offline option discovery proposed rencently. Can you explain the reason of choosing DDO?
[1] The Option-Critic Architecture
[2] A Compression-Inspired Framework for Macro Discovery

---

> ### Author Response · Authors · 2022-08-02
> **Response to Reviewer 94Fc**
>
> We thank the reviewer for the feedback and questions. We respond to each of the questions below and have included new empirical results, which have improved the paper. Please let us know if these responses have addressed the concerns.
>
> **Reviewer 94Fc asks why this work compares with DDO.** This work primarily compares with approaches that learn skills by maximizing the likelihood of the demonstrations, as this is the focus of the underspecification problem studied in this work. We specifically compare with DDO (and VTA), because it is a highly influential and well-known member of this group of maximum likelihood approaches. In addition, **we have added a comparison with Option-Critic**, one of the works referenced by the reviewer, on the ComPILE grid world navigation tasks. Using the implementation from https://github.com/lweitkamp/option-critic-pytorch, we find that LOVE significantly outperforms Option-Critic on both the easiest ($N_\text{pick} = 3$ (Dense)) and hardest ($N_\text{pick} = 5$ (Sparse)) settings. In $N_\text{pick} = 3$ (Dense), Option-Critic reliably achieves a reward of 2, whereas LOVE achieves a maximal reward of 3. On $N_\text{pick} = 5$ (Sparse), Option-Critic fails to achieve any reward, while LOVE achieves a high reward at ~0.7. We include training curves for Option-Critic in the updated Appendix I.
>
> |            | N_pick = 3 (Dense) | N_pick = 5 (Sparse) |
> |------------|--------------------|---------------------|
> | Option-Critic | 2.0                | 0.0                 |
> | LOVE (ours) | **3.0**               | **0.7**                 |
>
> *Maximum returns achieved by Option-Critic and LOVE.*
>
> **Novelty**. Reviewer 94Fc correctly notes that several prior methods encourage options to act for more timesteps or prevent skills from switching too frequently, which can similarly help avoid degenerate solutions like LOVE. However, these methods do not necessarily yield skills that help learn new tasks: e.g., longer skills are not always better for learning new tasks than shorter skills. Instead, the key hypothesis of this work is that the skills most relevant for new tasks are those that maximally extract common structure from the data, and LOVE provides a new compression objective that does so. The maximum of this objective achieves the information-theoretic limit of compressing the trajectories, which is not done in other works.
>
> Additionally, we have added **new empirical comparisons** with the below prior methods that regularize the switching or length of an skill:
> - *Entropy*: This regularizes the entropy of skills akin to [3], which they call parsimony.
> - *Num switches*: This penalizes switching between skills, following [4], a variant of the Option-Critic framework, suggested by the reviewer.
> - *VTA* ($N_{\text{max}}, l_\text{max})$: VTA includes a prior on its boundary variables, which effectively sets the maximum number of skills per episode to be $N_\text{max}$ and sets the maximum number of actions a skill can take to $l_\text{max}$. We experiment with several settings of $N_\text{max}$ and $l_\text{max}$, including those that use prior knowledge about the domain not used by LOVE.
> We compare these approaches with LOVE on the grid world navigation segmentation component (detailed in Table 2 of the submission). We find that LOVE achieves significantly higher F1 than these prior approaches.
>
> |           | entropy | num switch | VTA (3,5) | VTA (3,10) | VTA (5,5) | VTA (5,10) | Love (ours) |
> |-----------|---------|------------|-----------|------------|-----------|------------|---------------|
> | precision | 0.26    | 0.80       | 0.34      | 0.40       | 0.34      | 0.32       | **0.90**        |
> | recall    | 0.53    | 0.93       | 0.46      | 0.60       | 0.50      | 0.43       | **0.94**        |
> | F1        | 0.35    | 0.86       | 0.39      | 0.48       | 0.40      | 0.37       | **0.92**        |
>
> We note that the work [2] mentioned by the reviewer learns open-loop skills that do not depend on the state, and leaves learning closed-loop skills for future work. This cannot be directly applied to our setting, since solving the grid world navigation tasks requires conditioning on the state. It is also worth noting that extending [2] to condition on the state is non-trivial since the LZW algorithm used by [2] can only operate on discrete sequences (namely the action sequence and not the states).
>
> **Reference**
>
> [3] Discovering motor programs by recomposing demonstrations. Shankar et al.
>
> [4] When Waiting is not an Option: Learning Options with a Deliberation Cost. Harb et al.

---

> ### Author Response · Authors · 2022-08-05
> **Follow-up**
>
> Hi reviewer 94Fc,
>
> We wanted to follow up to see if our response has addressed your concerns or if you have any further questions. Thank you!

---

> ### Comment · Reviewer_gYjp · 2022-08-06
> **Follow follow-up**
>
> Dear reviewer 94Fc,
> We wanted to follow up to see if the authors' response has addressed your concerns or if you have any further questions. Thank you!

---

> ### Comment · Reviewer_94Fc · 2022-08-07
> **Good response**
>
> Thanks the author for the rebuttal, my concerns are resolved and I will improve the score.

---

### Official Review · Reviewer_cwAV · 2022-07-11

**Rating:** 7
**Confidence:** 4
**Soundness:** 4 excellent
**Presentation:** 4 excellent
**Contribution:** 3 good

**Summary:**

This paper presents a method from offline demonstration learning to HRL. The paper builds upon the generative model of VTA and add an information cost to enforce a better segmentation, which ends up aligning with the ground truth boundaries, The recovered options can be used in the HRL with augmented action space which shows good performance,

**Questions:**

Can you provide the comparison between your method and compILE?
Can you also provide some experiments on robotics task with continuous state/action space like jaco pinpad in [1] or reacher?

[1] https://arxiv.org/pdf/2103.10972.pdf

**Ethics Review Area:**

["I don’t know"]

**Limitations:**

Not that I can see.

**Strengths And Weaknesses:**

Strengths:
The motivation is clear, and the paper is well written and easy to follow. I find the proposed method to be novel and clean. The experiment results also suggest the method is effective at recovering the ground truth boundaries. The connection to MDL also seems convincing. The experiment on HRL and learning new tasks is also impressive.


Weakness:
I would like to see discussion and comparison with CompILE [1], which I think is quite similar to this work, since they are both based on generative models.


[1] https://arxiv.org/abs/1812.01483.

---

> ### Author Response · Authors · 2022-08-02
> **Response to Reviewer cwAV**
>
> We thank the reviewer for the encouraging feedback.
>
> We completely agree that **ComPILE would be a valuable additional point of comparison**. Unfortunately, we are unable to directly compare with ComPILE, as the only publicly available implementation does not support control (i.e., it cannot take actions and perform RL tasks). However, we have attempted to provide some imperfect comparisons in two ways:
> - First, we note that ComPILE belongs to the category of approaches that learn skills by maximizing the likelihood of the demonstrations, but with **additional assumptions of knowing the number of skills present in each demo**. Our work considers the setting without this assumption, as the optimal decomposition of the demos is often unknown a priori, and without this assumption, ComPILE would be equivalent to VTA with a different graphical model. Hence, the comparisons with VTA may serve as an imperfect comparison with ComPILE.
> - Second, we directly compare the grid world segmentation results reported in the ComPILE paper with ours from LOVE below. We note that this comparison is imperfect, as we use a custom implementation of the environment since ComPILE’s implementation is not publicly available, but it may still provide some rough signal. We find that LOVE performs comparably to ComPILE on the segmentation task, even though ComPILE leverages the strong assumption of knowing the number of skills present in each demo, which is not used by LOVE. When this information is not provided to ComPILE, its performance can significantly degrade (e.g., see Table 3 of [1]).
>
> |                   | ComPILE | Love |
> |-------------------|---------|------|
> | Boundary Accuracy | 1.0*    | 0.98 |
> | Reconstruction    | 1.0*    | 1.0  |
>
> \* *estimated from the bar plot in Figure 7 of ComPILE since the paper does not provide exact numbers.*
>
> Furthermore, we have added new comparisons with Zhang et al. [2], and Option-Critic [3], as requested by other reviewers in Appendix I and J respectively. LOVE significantly outperforms both.
>
> Finally, we agree that continuous control tasks could further strengthen this work. Our code is built on a framework specifically designed for discrete control, and the discussion period is not long enough to implement these. We plan to include continuous control tasks in the next version of the paper.
>
> **Reference**
>
> [1] CompILE: Compositional Imitation Learning and Execution. Kipf et al.
>
> [2] Minimum description length skills for accelerated reinforcement learning. Zhang et al.
>
> [3] The Option-Critic Architecture. Bacon et al.

---

### Official Review · Reviewer_CrHt · 2022-07-11

**Rating:** 7
**Confidence:** 3
**Soundness:** 3 good
**Presentation:** 3 good
**Contribution:** 3 good

**Summary:**

This submission introduces a general-purpose prior for option discovery from demonstrations: minimum description length, i.e., a compression objective. The authors extend the VTA model from Kim et al., 2019, to include a compression objective and a policy decoder (which can constitutes the option policies). The experiments demonstrate that the algorithm is effective in discrete action settings with both 2D and 3D observations.

**Questions:**

See above

**Limitations:**

The conclusion in the main body mentions possible future work but fails to raise awareness on limitations, or at least on open (and as for now) unaddressed open questions. For example, would the proposed method potentially work on the maze navigation tasks in D4RL (https://sites.google.com/view/d4rl/home)? I think that corresponding evaluations can be regarded as outside the scope of this work, but if the authors see possible issues in applying the method to other popular offline RL tasks, this would be an interest addition to the paper's conclusion.

**Strengths And Weaknesses:**

The paper is well-written, and the motivation and presentation is good, and I think that finding good, general priors for option discovery is of high interest. The experiments are described clearly and, although they clearly expose a very suitable structure for learning discrete options, showcase the efficacy of the method sufficiently well.

As the authors note, MDL for option discovery was recently proposed by Zhang et al., 2021 -- it would have been interesting to include this as another baseline for comparison.

In terms of ablations, I'd encourage the authors to further analyze the effective numbers of skills that are used (explaining the effect of filtering unused skills as described in section 5). In a similar vein, I was wondering how robust the proposed algorithm is to the number of options that should be learned. If I understand correctly, in the experiments the number of objects to navigate to is known beforehand and the number of options is specified accordingly. Suppose that this information is hard to obtain from a given set of demonstrations -- how much would it hurt to specify too many or too few options? To add to that, in Fig. 8 (Appendix), 7 out of 10 skills seem to result in the same actions. Why is that?

---

> ### Author Response · Authors · 2022-08-02
> **Response to Reviewer CrHt**
>
> We thank the reviewer for the encouraging feedback and suggestions for additional experiments, which have improved the paper’s clarity. We clarify the reviewer’s questions and include the suggested additional experiments below, and have updated the paper accordingly. Please let us know if these responses have addressed the concerns and we would be happy to answer any further questions.
>
> 1. **We have added comparison with Zhang et al., ‘21, requested by reviewer CrHt.** We note that Zhang et al., ‘21 learn open-loop skills, which do not condition on the state and therefore cannot adapt to different states. We also emphasize that the MDL used by Zhang et al., ‘21 is equivalent to the variational inference (VI) used by VTA, which can be seen as greedily compressing each skill independently, rather than compressing a whole trajectory as LOVE does. The table below reports segmentation results on the Color domain and grid world, akin to Tables 2 and 3 in the paper, as the response period does not leave enough time to run the downstream RL component. Zhang et al., ‘21 performs the same as VTA on the Color domains, as they use the same VI, and there is no state, which achieves lower precision / recall than LOVE. On the grid world navigation, Zhang et al., ‘21 fails to recover the boundaries, unlike LOVE, as skills that navigate to objects require observing the state. These results are included in Appendix J of the updated paper.
>
> |           | LOVE | Zhang et al., '21 |
> |-----------|------|-------------------|
> | Precision | 0.99 | 0.87              |
> | Recall    | 0.85 | 0.78              |
> | F1        | 0.91 | 0.82              |
>
> *Results on SimpleColors. Precision, recall and F1 are measured between the predicted task boundaries and true task boundaries (higher is better).*
>
> |           | LOVE | Zhang et al., '21 |
> |-----------|------|-------------------|
> | Precision | 0.99 | 0.84              |
> | Recall    | 0.83 | 0.82              |
> | F1        | 0.90 | 0.83              |
>
> *Results on ConditionalColors.*
>
> |           | LOVE | Zhang et al., '21 |
> |-----------|------|-------------------|
> | Precision | 0.90 | 0.79              |
> | Recall    | 0.94 | 0.34              |
> | F1        | 0.92 | 0.48              |
>
> *Results on grid world navigation.*
>
> 2. **Skill number ablation** These are great questions. The number of pre-filtering skills $K$ is not specified beforehand according to the number of objects to navigate to, and the number of objects to navigate to is not used in any way to set LOVE’s hyperparameters. Conceptually, $K$ can be set conservatively: LOVE requires a minimum value of $K$ to cover the behaviors in the demonstrations, but LOVE can gracefully prune out excess skills if $K$ is too large in the filtering step described in line 213 of the paper. We empirically validate this intuition by varying the value of $K$ as suggested by the reviewer on the segmentation component of the ComPILE grid world navigation environment, for $K = 2, 5, 10, 15, 20, 30, 50$ (we use $K = 10$ in the paper). Performance degrades when $K$ is set too small (e.g., $K = 2$ and $K = 5$), but performance remains high across a wide range of larger $K$ values.
>
> |           | K = 2 | K = 5 | K = 10 | K = 15 | K = 20 | K = 30 | K = 50 |
> |-----------|-------|-------|--------|--------|--------|--------|--------|
> | precision | 0.27  | 0.80  | 0.90   | 0.96   | 0.96   | 0.95   | 0.95   |
> | recall    | 0.53  | 0.91  | 0.94   | 0.96   | 0.95   | 0.96   | 0.93   |
> | F1        | 0.35  | 0.86  | 0.92   | 0.96   | 0.95   | 0.95   | 0.94   |
>
>    *Results on different values of $K$, the number of skills.*
>
> 3. **Why do some of the skills in Figure 8 appear redundant?** As mentioned in lines 918-920 in Appendix H, the skills visualized in Figure 8 are those before the automatic filtering step. After filtering, only 7 skills remain (skills 1, 2, 4, 5, 6, 8, 9), which exhibit less redundancy. Additionally, note that while several of these skills still pick up the same object in the specific state visualized in Figure 8, **they act differently from each other in other states**, which can be seen in Figure 9. We have updated the last paragraph of Appendix H to make these points clearer.
>
> 4. **Reviewer CrHt suggests adding discussion on open problems, specifically learning skills from offline data beyond demonstrations.** We have added the following discussion to the conclusion. In general, LOVE applies when there are useful structures that can be extracted from multiple trajectories. This is often present in multi-task demonstrations, which solve related tasks in similar ways but could be extended to general offline data. However, an open future challenge to adapting to general offline data is to ensure that the learned skills do not over-focus on potentially noisy or unhelpful behaviors often present in offline data.

---

> > ### Author Response · Authors · 2022-08-09
> > **Have the new experiments addressed concerns?**
> >
> > Dear Reviewer CrHt,
> >
> > Thank you for the suggestions for improving the paper. Our above response includes two additional experiments to address questions raised in the review, experiments that we believe further strengthen the paper. Together with the discussion above, **have all the concerns been addressed?** If not, we would be happy to continue the discussion and/or revise the paper.

---

### Author Response · Authors · 2022-08-03
**Author Response Summary**

We thank all reviewers for taking the time to provide valuable feedback. We have responded to each reviewer separately, and have incorporated their feedback to strengthen the paper, including requested additional experiments. Please let us know if there are any remaining concerns or questions. Below is a summary of the main changes to the paper:
- **Additional comparisons.** We have included several new points of comparison suggested by the reviewers, specifically 1) Option-Critic; 2) three other approaches that regularize skill length or switching frequency; and 3) Zhang et al., ‘21.
- **Pre-filtering skill ablations.** We have included new experiments that vary the number of skills LOVE uses before filtering.
- **Open problem discussion.** We have added discussion on the open problem of applying LOVE to general offline data beyond demonstrations, or to online skill learning.
- **Clarifications about novelty.** We have clarified the difference between LOVE’s compression and compression used in other works, such as [1, 2, 3], which is that LOVE provides a principled fully-differentiable information-theoretic objective that enables learning closed-loop skills, whereas prior compression only compresses action sequences, which yields open-loop skills that do not condition on the state.

[1] A Compression-Inspired Framework for Macro Discovery. Francisco M. Garcia, Bruno C. da Silva, Philip S. Thomas.

[2] Augmenting Policy Learning with Routines Discovered from a Single Demonstration. Zelin Zhao, Chuang Gan, Jiajun Wu, Xiaoxiao Guo, Joshua B. Tenenbaum.

[3] Minimum Description Length Skills for Accelerated Reinforcement Learning. Jesse Zhang, Karl Pertsch, Jiefan Yang, Joseph J Lim.

---

### Meta-Review · Area_Chair_UVgV · 2022-08-24

**Recommendation:** Accept
**Confidence:** Certain

**Metareview:**

This paper studies the problem of learning options in multi-task reinforcement learning. The authors note that previous works optimize an underspecified objective and they propose adding an extra term to the objective function that relates to the description lengths of skills. The authors study their approach and show empirically that it scales to high-dimensional problems and performs well compared to previous approaches. The discovered skills can also be used to solve new tasks using fewer samples than previous approaches.

The initial reviews were overall very positive for this paper. During the author-reviewer discussion, the authors provide additional results and provided satisfying answers to most reviewer comments. As a result, the reviewers are unanimous that this work should be accepted and the discussion period did not reveal any other elements to report here.

I am pleased to recommend acceptance, congratulations! It seems like the current version of your manuscript already addresses most if not all of the points raised by reviewers. In addition, please do not forget to discuss the limitations of your current work including e.g. different domains that it might (not) work in (see the comment from reviewer CrHt).


**Award:**

No

---

### Decision · Program_Chairs · 2022-09-14

Accept